# What Is the Value of a Global Health Research School for PhD Students?

**DOI:** 10.3390/ijerph192013361

**Published:** 2022-10-16

**Authors:** Elisabeth Darj, Irene Bisasso Hoem, Elin Yli Dvergsdal

**Affiliations:** 1Department of Public Health and Nursing, Norwegian University of Science and Technology—NTNU, 7034 Trondheim, Norway; 2Department of Education and Lifelong Learning, Norwegian University of Science and Technology—NTNU, 7034 Trondheim, Norway; 3MH Administration, Faculty of Medicine and Health Science, Norwegian University of Science and Technology—NTNU, 7034 Trondheim, Norway

**Keywords:** alumni, PhD, education, global health

## Abstract

The aim of the study was to investigate previous PhD students’ views on the Norwegian Research School of Global Health and its activities. Of the research schools for PhD students, few focus on global health and even fewer have evaluated the students’ perspectives of the schools. In this study, a questionnaire including quantitative and qualitative questions were sent to alumni PhD students. Demographic status was investigated, along with the alumni’s views on activities offered at the research school, suggestions for improvement, views on his or her social life as a student and as a member of the school. A total of 60 alumni were contacted by email and invited to participate in an anonymous online survey. The response rate was 65%; 90% were in employment and a few were seeking employment. All research school activities were evaluated as useful. Content analysis of qualitative questions generated three categories of the alumni’s reflections on their involvement in the research school: valued activities, challenges, and future. The alumni expressed a wish for continued contact with the school. The findings indicated that a research school for PhD students with similar interests should be continued; although modifications should be considered, based on the specific challenges revealed in the evaluation.

## 1. Introduction

As a measure to improve the education of post-graduate students, an increasing number of research schools have become established in various specific fields. Their aim is to support students’ learning, to inspire research and to build national and/or international networks. Funding such schools is expensive; thus, it should be mandatory to evaluate the value of the outcome of the school. A review, from 2002, of studies on after-school programs concluded that evaluations of programs were based on research designs that were too limited to support causal conclusions, and information available was insufficient to allow for meta-analysis [1]. Information in the literature regarding PhD alumni’s evaluations of research schools is scarce and difficult to find. One study of a global health program report from 84 medical trainees in various specialties, revealed that 95% would recommend the program, 84% felt more prepared to deliver global health care, and 78% reported career plans that included global health. This study concluded that such a health education program across specialties could be feasible and effective [2]. However, a group of medical trainees are not directly comparable with a group of international PhD students. Monitoring and evaluation of programs is necessary to determine their progress and make adjustments for a better outcome in the future. 

Global health is an increasingly important topic. This has been extremely obvious during the current worldwide COVID-19 pandemic [3,4]. Therefore, well-educated clinicians and researchers are needed and when they enroll in a research school, they should be able to be confident that the school’s activities are of high quality. PhD students studying global health at universities around the world are focusing on various factors that affect health in relation to several of the UN’s Sustainable Development Goals, such as climate change, areas of conflict, inequities, poverty, migration, infections, zoonoses transferred to humans, and lack of health services [5]. International health has been used to describe the provision of health care or development from a wealthy country to a poorer region, for example, low- and middle-income countries (LMICs). [6]. The goal of global health, as described by Kickbusch, is the ability to achieve equitable access to health in all regions of the globe, a part of public health. The academic global health community is now focusing more on health equity and reexamining the dynamics of global health education and partnerships [7]. Health risks transcending national boundaries and governments, call for joint actions by policymakers and international organizations, which have shown their indisputable importance during the current pandemic. Consequently, education of new researchers is important, as are the methods by which knowledge should be effectively transferred and implemented. The education of new researchers must include a variety of topics, such as medical science, research methodology, innovation and technology, and environmental and behavioral sciences. In Norway, PhD students who come from different countries have differing backgrounds; similarly, the teachers have different types of expertise. 

The Norwegian government and universities have a vision to increase internationalization. In 2015, the Research Council of Norway (RCN) launched a call for research schools [8]. The universities of Trondheim, Bergen, Oslo, and Tromsø, together with the Norwegian Institute of Public Health, decided to apply jointly for funding for a national research school in global health. Norway is a small country, with 5.5 million inhabitants, and the five institutions anticipated that by acting together, a synergetic effect would make the outcome of education stronger than from each of them separately [9]. The main aim of the school is to strengthen the quality of PhD education in global health by providing relevant activities. The proposal was funded by the RCN for six years and the research school enrolled its first PhD students as members in September 2016 [8]. 

The board of the newly established research school comprised a director, a chairperson, one representative senior researcher from each of the partner universities, an international representative, and two PhD students elected as representatives by their fellow PhD students [10]. Eligible students accepted for membership in the research school have to be registered at a Norwegian university or an institution for higher education and studying a topic concerning global health. A website was developed and is regularly updated on how to apply for membership, funding options, relevant events, and courses [10]. Since 2017, an annual PhD conference has been arranged, with invited guest lecturers and a focus on various transferable skills, such as writing, oral presentation, and posters, during which the students receive feedback from senior researchers and from their peers. 

As the students have different personalities and integrate knowledge in different ways, diverse educational skills and learning methods are applied in the courses and presented in the conferences, including visual, audial, read/write, and kinesthetic methods (VARK) [11]. Through mentoring, feedback and training in critical thinking, and improvement in the students’ academic skills, can be achieved, as described in Bloom’s taxonomy [12]. The program also made use of technology, in line with findings by Perkins et al., that even though there is limited data available on new methods used for assessment, and no one “best approach” exists for evaluation and feedback, a multimodal approach that is based on technology can be effective and aid in guiding programs. [13]. The research school received financial support to develop new courses, as well as mobility grants to support students taking courses at other Norwegian universities. Each of the universities provided different courses, some of which could be relevant for students attending universities in other cities; without financial support for travel and attendance, these courses would have been inaccessible to many students. Support was available for students who wanted to attend an international institution to acquire specific skills, or to participate in a conference, for example, through instruction in writing abstracts. Local workshops at each university were supported, as well as courses, targeting teachers, that were designed to improve their supervision skills. 

In 2018 and 2020, two regional supervision meetings were planned to be held in Africa and Asia. A workshop in Uganda attracted 20 local supervisors from partner universities in East Africa, but the meeting in Asia had to be postponed due to the COVID-19 pandemic. 

Between the time when the research school opened in 2016 and when the survey was conducted in 2020, a total of 209 members had been enrolled. Of the 209 members, 60 had defended their thesis, 150 (72%) were active students and 7 students had left the school before defending their thesis. 

All courses were evaluated with questionnaires directly after each course. A strength of such a formative participatory evaluation made during an ongoing study is that it is possible to act based on the result [14]. The main aim of this national research school is to strengthen the quality of PhD education in global health by providing relevant activities. Therefore, we approached the PhD alumni to investigate their views on their entire research school experience. Such an evaluation had not been conducted previously. Thus, we anticipated that our results would help the board of the school to adjust and improve the provided education. The aim of this study was to perform an investigation among alumni on their views on the research school targeting global health, the provided activities, and whether they found it valuable for their future careers.

## 2. Material and Methods

### 2.1. Study Participants

Previous members of the NRSGH, who had finalized their doctoral research and defended their thesis, were invited to participate in the study by answering an anonymous questionnaire online. All participants had been registered at a Norwegian university. After they had been awarded their PhDs, they were living in many different places worldwide. 

### 2.2. Study Design

At the end of 2020, a survey with both quantitative and qualitative questions was sent to all eligible alumni. A reminder was sent out in January 2021. Quantitative questions regarding gender, membership, activities, and work status after completion of their education were included. To gain a better understanding of the former students’ views on the research school, the survey also included open-ended qualitative questions about suggestions for improvement, possible new courses, and the social situation at the school, as well as any other information that the participants chose to relate. The aim of the survey was to capture individual perceptions and experiences to generate data that reflected the students’ views. The alumni were accustomed to answering similar questionnaires as they had previously given feedback at the end of each of the research school’s activities. 

### 2.3. Data Collection

A SurveyMonkey^®^ (Momentive Inc., San Mateo, CA, USA) questionnaire was sent to the most recent email address of each of the former students, with one reminder. The anonymous answers could not be traced back to an individual. 

### 2.4. Data Analyses

The quantitative data were described frequencies. Inductive content analysis of the qualitative data was performed, whereby meaning units were searched for in the text material, condensed and coded. The coded units were merged into categories and subcategories in accordance with the qualitative analysis method described by Graneheim and Lundman [15]. In this article, the findings are illustrated by quotes from the participants. 

## 3. Results

A total of 60 alumni were eligible to participate in the study and 39 answered the questionnaire, giving a response rate of 65% after one reminder. Among the former students, 17 were women (43.6%) and 22 men (56.4%). Two students had registered in 2017, 10 in 2018, 14 in 2019, and 13 in 2020. They had been members of the research school for two and a half years on average. To avoid any risk of disclosure, participants were not asked to name the university at which they had been registered. 

In 2020, 35 (89.7%) of these alumni were working and 4 (10.3%) were still seeking employment, and 23 (59%) had permanent employment. All jobseekers had defended their PhD thesis in 2020 (Table 1). 

The participants were asked to evaluate their membership of the research school and activities in which they had participated, based on four options: very useful, useful, less useful, or not applicable. All 39 participants evaluated their membership of the research school as very useful, 32 participants evaluated all activities combined as very useful or useful and 6 indicated that the activities question was not applicable to them. Among the specific activities, the provision of travel grants was the most appreciated activity, followed by the provision of international training grants, mentoring and feedback (Table 2). 

During the analysis of the open-ended questions, three main categories were generated: *valued activities*, *challenges*, and *future*. These were subsequently subdivided into seven subcategories (Table 3). 

### 3.1. Valued Activities

The activities were highly valued by the participants. The participants wished for extra time for the conferences, and more courses, as those activities also enabled them to access a network of scholars. 

#### 3.1.1. PhD Conferences

It was obvious that the PhD conferences were perceived as important and valuable. The conferences were mentioned as being beneficial, useful, a source of new academic skills, and an opportunity for students to reflect on their own research. Furthermore, the PhD conferences gave students from various countries outside Norway a sense of community. They appreciated the mentoring and feedback given by senior researchers and by their peers. They also appreciated the possibility to attend virtually if they were not in Norway at the time and during the pandemic. However, one former student believed that, due to heavy workloads, it could be hard to get permission to attend activities outside their own university:

“PhD conferences are the most important to keep and further develop. They made me very useful in various ways, such as skills in academic writing and administrative skills.”

“I attended two conferences which were very useful for exchanging knowledge and strategies amongst PhD students and also learning more practical presentation and writing skills.”

#### 3.1.2. Courses

All universities in Norway have mandatory courses for PhD students, in, e.g., methodology, communication and ethics. The Norwegian Research School of Global Health (NRSGH) developed specific courses targeting global health. Those courses covered academic writing, focusing on how to present and submit articles relating to global health for publication, and an innovation course that emphasized solutions aimed at improving health, especially in resource poor regions. The alumni also valued a workshop on how to communicate research results outside academia: 

“I got a partnership. I acquired skills, motivation and progress.”

“[I wish] more on academic writing and communication of one’s research.”

Some participants specifically suggested the inclusion of courses to improve alumni’s teaching skills to transfer their newly acquired knowledge:

“You need to also include training in education teaching […] for me, the PhD is not the end of learning process. I need more.”

Support to attend courses and conferences, and travel grants, were scored highly in the quantitative scoring system, although not all alumni had taken advantage of those options when they were students. The specific activity of providing grants was not much discussed in the open questions and answers, other than being mentioned as beneficial.

#### 3.1.3. Networking

An important reflection was that networking and socializing with other students was highly cherished. When the alumni were students, they met peers with the same research interests and could discuss common challenges and opportunities, and they could connect with researchers from the same geographical regions. Some participants expressed that occasionally they had been alone, either when isolated during the pandemic or when travelling outside Norway for data collection, but their already established network with others was valued. One former student expressed the opinion that most of the activities benefitted students who were in Norway, and it would be beneficial if more activities were made digital and accessible to those outside the country and especially during the pandemic:

“Peer-to-peer interaction can enrich knowledge and skills. Courses were useful forums for exchanging experiences and lessons learned by other members.” 

“Networking across the globe [is needed], otherwise we feel so detached.”

### 3.2. Challenges

The category challenges comprised two subcategories: being in a new country and the special situation during the COVID-19 pandemic (see Table 3). Being a PhD student in a foreign country, as are many of the members in the research school, poses special challenges. Since our research for this article was performed during an ongoing pandemic, the special situation experienced by the students in 2020 was understandably an important issue to them. 

#### 3.2.1. Being New in the Country

Socializing with others was important to the students who participated in the survey and there was a wish for an introduction to how to act as a PhD student and how to balance an exceptional amount of work and a social life. The alumni suggested that to have been given an introduction by a previous student with a similar background would have made it easier for them to adapt to the new environment and to learn some Norwegian expressions, and it would have reduced the stress they had experienced and encouraged them to participate in joint outdoor activities: 

“It took me more than six months to feel I knew my way […] Involving basic Norwegian language courses for international fellows could help ease social stress.” 

Concerns regarding students’ economic situation and mental health were mentioned, as well as a need for the school to provide support in the form of counselling: 

“You also need to include other challenges faced by international PhD students concerning funding and quality of life […] especially mental health while studying.” 

Some participants mentioned their friends who had been awarded their PhD degrees from universities in other countries, where requirements for submitting a doctoral thesis differed from those in Norway. Therefore, the participants wanted a specific course on how to write their summary of their paper-based thesis and prepare for its defense in the Norwegian academic system.

#### 3.2.2. Special Situation during the COVID-19 Pandemic

There was a wish for more support during the pandemic, since when the survey participants were students, they were individually isolated, some in Norway and others in their home countries. Several of the participants suggested that students should be given the opportunity to take more courses online, also after the pandemic, and, thus, be able to gain the required knowledge regardless of where they are located:

“I was not involved in the research school for a long duration. I felt alone.”

“Explore ways to support PhD students during pandemics, such as COVID-19.” 

### 3.3. Future

Although many of the alumni had permanent or temporary work, the category future reflects their own need for continued learning and their suggestions for improvements in the research school. Two subcategories illustrate these perceptions: continued support and transition from student to employee (see Table 3). 

#### 3.3.1. Continued Support

It was apparent that the alumni wanted to have continued contact with the research school and participate in activities. They requested a continuation of support, for example, in the form of travel grants and financial support to participate in the PhD conferences, to keep in contact with others in their research field of global health, and the possibility to take courses online. Enrolled members have access to social media explicitly for them. The alumni wished to have continued access to information on the school’s website and to participate in a forum for alumni, where they could share experiences and findings with their former fellow PhD students and current colleagues:

“Consider the research school’s alumni for courses automatically, especially online.”

“Continued support for developing proposals and protocols.” 

#### 3.3.2. Transition from Student to Employee

The second subcategory, ‘Future’, indicates that provision of advice on career planning, specific courses for writing grant applications, information about job opportunities, more knowledge about the transition from being a PhD student to being employed, and information on how to access international organizations, industries and NGOs for mentorship and potential work were required: 

“More on career development, transition from the PhD to other careers.”

“Career guidance and networking across the globe”

## 4. Discussion

The research school with its specific interest in global health was valued and appreciated by the PhD alumni and deemed beneficial in helping them to accomplish their education, as well as to socialize and network. To a certain extent, the school also prepared them for a career after completion of their education. Nevertheless, alumni asked for more activities connected to the period after completion of their PhD education and as they started independent research. Membership and participation in the school’s activities were highly prized among the alumni. The membership of the school added value to the time they spent as PhD students in Norway, as the school specifically targeted their common topic, that of global health. 

Our interpretation is that the main aim of the research school has been fulfilled and the quality of the education provided for PhD students has been strengthened, specifically by targeting global health issues, providing relevant activities, facilitating recruitment of young researchers in global health, and broadening of their international network. In its mid-term evaluation of the school, the funding body, RCN, made the following encouraging comments: ‘*The NRSGH is an ambitious research school, working in a field that is more important now than ever. Therefore, training of young researchers for the future within global health is an important issue globally*’ [8]. The Norwegian Research School for Global Health intends to constantly seek improvement and our study has illuminated some of the challenges to consider in the remaining period of the school.

The ongoing COVID-19 pandemic affected the students. There were strong restrictions in place during 2020 in Norway and elsewhere, and, therefore, many courses and meetings were cancelled, and other meetings were held digitally. Although the situation was not ideal, teachers and students alike had to adjust to it. The alumni’s perception of isolation during the pandemic is in accordance with a study of students in general. In that study, all students experienced stressors during the pandemic, and those without pre-existing mental health concerns reported greater increases in social and academic isolation, relative to students with pre-existing mental health concerns [16]. During the pandemic, the use of various pedagogic techniques was limited and reduced to online lectures. Frankenhoff and Szczypińska discuss how people manage stress and they refer to Antonovsky’s concept of ‘Sense of coherence’ [17,18]. According to Antonovsky, the ability to adapt to a new process, to understand new circumstances, handle stressful situations, and to see purpose in behavioral change is essential [19]. Some teachers and students might not have relevant knowledge about how to deal with new educational ways or have the necessary environmental support or motivation to cope sufficiently well. However, the results of our survey revealed that the possibility to attend courses and be involved in activities online would be a way to open access for the students who are away from Norway or who do not have the opportunity to travel within Norway. During the travel restrictions, local meetings were arranged for students living in the same towns. The meetings were highly valued by the alumni and the idea could be initiated for PhD students doing data collection abroad, also after the pandemic. That would also enable the research school to keep in contact with students when they are away. Despite the challenges, the research school was appreciated, and the alumni wanted an opportunity for continued learning and contact.

The qualitative text material revealed that PhD conferences and courses were regarded as very helpful. By contrast, the quantitative scoring system revealed that the highest scores were for travel grants. However, some data were missing in the part of the survey relating to conferences and courses, which may reflect that several students did not have the opportunity to participate in those activities during 2020, although the students who had participated were satisfied with the opportunities offered.

Coming from a foreign country to Norway poses specific challenges for many members of the school. A workshop with an introduction to not only the academic world, but also Norwegian culture, was, thus, initiated in the research school’s activities in 2021. Another cause of stress, which may influence the students’ mental health, concerns their economic situation. It is unquestionably demanding for students to be away from their family and be worried about their economic situation. Although some websites provide information and encourage students to combine their PhD studies with work at the same time, the research school does not support this practice, as the courses are full-time and challenging [20]. Though none of the participants mentioned that they had undertaken paid work while studying, some might have done so.

A gender imbalance relating to employment status was evident in our survey results, as more men than women had secured permanent employment and more women had temporary jobs after being awarded their PhD. This finding may be problematic, yet, despite a small sample size, it may reflect results from other studies that show that women do not advance as far in their academic career compared with men [21]. Female mentoring is seen as a helpful career tool to assist with strategic planning for the future [22]. However, this issue was not mentioned by our study participants.

The survey findings indicate that the research school needs to consider the provision of support for students in career planning, their transition from a student to an employee, and how to secure a permanent job, as well as whether, or how, continued contact with the school could be provided. Starting in 2021, career planning is offered in specific workshops by the Norwegian Research School of Global Health. Other responsibilities, such as counselling on mental health and how to prepare for the PhD defense, rest more on the individual partner universities and not on the research school.

### Strengths and Limitations

One limitation of our study is that the sample of participants was small, and provided only a description of the material, which affects our ability to draw conclusions from the quantitative results. However, using quantitative data, complemented by qualitative data, illustrated with quotes, gave us a deeper understanding of the previous students’ thoughts of the value of participating in the research school. Both the qualitative and quantitative data indicated similar experiences, thus strengthening our conclusions.

A further consideration is that two of the authors are engaged in the research school and, therefore, may have preconceptions about how students perceive their participation in the school. To compensate for this, a researcher from another faculty was invited to co-author this article. Furthermore, we have thoroughly described the context, methods, and systematic analyses of the data, and used relevant quotes from the participants to ensure and emphasize the trustworthiness regarding credibility, transferability, and dependability. The qualitative part of our study provided insights into former students’ perceptions of the research school, including the school’s advantages and challenges. Consequently, adjustments have been made. However, individual interviews might have afforded a deeper understanding of the alumni’s perceptions. 

## 5. Conclusions

The Norwegian Research School of Global Health has contributed to building a new generation of researchers. Through its collaboration with international partner institutions, the school has strengthened the school’s research networks. The quantitative results gave us information that indicated all students highly valued their membership of the research school. The majority appreciated the activities, grants, conferences, and networking. However, adding the qualitative part gave more insight into their thoughts. That it was quite challenging to come to Norway was not covered by any of the questions asked in the online survey. The qualitative part also allowed alumni to reveal their feelings of isolation and need for more support during the COVID pandemic. The alumni requested support also in the future and they wished to have access to the research school’s activities after they had finished their PhD work. The board of the school could later act upon these requests. We conclude that the main aim of this national research school, to strengthen the quality of PhD education in global health by providing relevant activities, was fulfilled. Furthermore, data from this study reflect the students’ views that global health education in the regime of the research school was appreciated and highly valued. Therefore, taking into consideration the results we have presented, we propose that such a research school should continue and securing finances for a comparable educational program would be of substantial value. There is a need for the government to continue financial support, and for universities to join and make use of the synergy of providing courses together and working in partnership, and, furthermore, to offer slightly different PhD courses and allow students to take courses approved at other universities. The Board have additionally discussed a similar Nordic collaboration.

## Figures and Tables

**Table 1 ijerph-19-13361-t001:** Description of the work situation of the PhD alumni (N = 39) from the Norwegian Research School of Global Health.

Employment Status	Women (n = 17)		Men (n = 22)		Total (N = 39)	
	n	(%)	n	(%)	n	(%)
Permanent employment	7	(41.2)	16	(72.7)	23	(59.0)
Temporary employment	8	(47.1)	4	(18.2)	12	(30.7)
Jobseeker	2	(11.8)	2	(9.1)	4	(10.3)

**Table 2 ijerph-19-13361-t002:** Usefulness of activities evaluated by PhD alumni of the Norwegian Research School of Global Health (N = 39).

Activity	Very Useful	Useful	Less Useful	Not Applicable	Missing Data
Membership of the research school	39	0	0	0	0
All activities	22	10	0	6	1
Provision of travel grants	21	7	0	11	0
Provision of international training grants	15	5	1	18	0
Mentoring and feedback	12	12	3	12	0
Courses	10	11	9	0	9
PhD conferences	7	13	0	9	10

**Table 3 ijerph-19-13361-t003:** PhD alumni’s views on aspects of membership of the Norwegian Research School of Global Health.

Categories	Subcategories
Valued activities	PhD conferences
	Courses
	Networking
Challenges	Being new in the country
	Special situation during the COVID-19 pandemic
Future	Continued support
	Transition from student to employee

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
