# Peer review of "What Is the Value of a Global Health Research School for PhD Students?"

_ijerph, 2022, doi:10.3390/ijerph192013361_

Round 1

Reviewer 1 Report

1.Title: Please rethink of the title. This is not a Title but the answer to the research already. We do not answer the questions before doing research

2.Abstract is wordy. I wonder why the author/authors indicated the titles like Objective, Methods, Findings etc. I expected a written exposition which is sequential and at the sometime systematic.

3.I wanted to know why the authors chose Mixed Methods instead of just qualitative or quantitative?

4.This is the linear sequence I expected: Statement of the Problem =Yes, although I may need a lot more in scientific language

 -The topic=Yes, it is available

- The research problem=Not available

 -Background and justification (philosophical view points) = Why mixed methods paradigm? Not clear here

 -Deficiencies in the evidence=It is not clear from your research

- Audience =Yes, it is clear

- Definition of Terms= Did you define?

5.IRB process= There is only a brief statement about IRB. I wanted to hear more on how IRB was followed step by step. And how and how long are they going to store the data after this.

6.On the Results section-=How do the qualitative (qual) findings explain (expand on) the quantitative (QUAN) results?

7.Conclusion:

Your conclusion should match the findings and explain well why this research was important/viable and what is it contribution in academia. Please rethink of the summary which match well as this looks like misplaced.

8.Reference section: Are not in APA format-Not even alphabetically written

9. My comments:  This research however can be good for scholarship if revisions are done extensively clarifying different components mentioned above.

Author Response

Reviewer 1.

Thank you for your valuable comments, which helped us significantly in improving the manuscript. Please find our answers in red, below your comments.

1.Title: Please rethink of the title. This is not a Title but the answer to the research already. We do not answer the questions before doing research

Thank you for this comment. ‘A research school targeting Global Health is highly valued by its PhD alumni’ has now been changed to: ‘What is the value of a Global Health research school for PhD students?’: page 1, line 1-4

2.Abstract is wordy. I wonder why the author/authors indicated the titles like Objective, Methods, Findings etc. I expected a written exposition which is sequential and at the sometime systematic.     We agree with your suggestion and have now changed the abstract, there are no more headings in the abstract. It reads now as follows: page 1.

The aim of the study was to investigate previous PhD students’ views on the Norwegian Research School of Global Health and its activities. Of the research schools for PhD students, few focus on global health and even fewer have evaluated the students’ perspective of the schools.  In this study, a questionnaire including quantitative and qualitative questions were sent to alumni PhD students. Demographic status was investigated, along with the alumni’s views on activities offered at the research school, suggestions for improvement, views on their social life as students and as members of the school. A total of 60 alumni were contacted by email and invited to participate in an anonymous online survey. The response rate was 65%; 90% were in employment and a few were seeking employment. All research school activities were evaluated as useful. Content analysis of qualitative questions generated three categories of the alumni’s reflections on their involvement in the research school: valued activities, challenges, and future. The alumni expressed a wish for continued contact with the school. The findings indicated that a research school for PhD students with similar interests should be continued; modifications should be considered based on the specific challenges revealed in the evaluation.

3.I wanted to know why the authors chose Mixed Methods instead of just qualitative or quantitative?

Thank you for this relevant comment. We wanted to know more about the material, the alumni as a group, and if the quantitative and qualitative answers pointed in the same direction. But we have now omitted the expression ‘mixed methods’ in describing the study design, both in the abstract and the main text, and we have used the terms quantitative and qualitative instead. We understand that the sample size of 60 alumni, is a description of the material. No power-calculation was performed, and this makes it impossible to generalize the outcome. However, using some quantitative data, which were complemented by qualitative data, illustrated with quotes, gave us a deeper understanding of the previous students’ answers. Both the qualitative and quantitative data indicated similar experience which strengthens our conclusions. Page 2,5

4.This is the linear sequence I expected: Statement of the Problem =Yes, although I may need a lot more in scientific language

We agree with the reviewer, we have included more about the necessity to evaluate a postgraduate program, in order to adjust what is needed for a better outcome. We have further incorporated more references from the literature. In the Introduction, we refer to a study which concludes that there is insufficient information available to allow for a meta-analysis of post graduate schools evaluations (Scott-Little, C et al 2002). We found one study which assessed a global health program, however the program was for medical trainees, which is not directly comparable to our study group, as the PhD students come from various disciplines (Morgan J et al 2020). We also refer to Perkins et al 2020, who states that there is no "best approach" for evaluation and feedback. However, a multimodal approach can be effective and aid in guiding programs. Page 2,3

-The topic=Yes, it is available

Thank you for the comments.

 - The research problem=Not available

The research problem is now stated more clearly with more information, page 4

 -Background and justification (philosophical view points) = Why mixed methods paradigm? Not clear here

Thank you. Regarding ‘mixed methods’, please see also the answer above. We have omitted the use of the expression ‘mixed method’ in describing the study design. We understand that the sample size of 60 alumni, is too small to generalize the outcome, and for further statistical analyses. However, using some quantitative data to describe the material, which were complemented by qualitative data, illustrated with quotes, gave us a deeper understanding of the alumni’s answers, in a kind of triangulation. Page 2,5

 -Deficiencies in the evidence=It is not clear from your research                                                                  We mean the deficiencies or lack of evidence is the lack of research on alumni’s views of the research school have not been investigated or known, as this is the first evaluation of the specific research school of global health. Page 4, line

- Audience =Yes, it is clear

Thank you.

- Definition of Terms= Did you define?

No, sorry we did not define specific terms we used.

5.IRB process= There is only a brief statement about IRB. I wanted to hear more on how IRB was followed step by step. And how and how long are they going to store the data after this.

The Institutional Review Board (IRB) of the Research Committee of Norway is established to protect the rights and welfare of human research subjects. They review all research proposals involving human participants, before starting any study. If they do not find any questions regarding personal health or integrity in the protocol, the IRB have not a duty to approve. The study followed the Declaration of Helsinki. Page 11

6.On the Results section-=How do the qualitative (qual) findings explain (expand on) the quantitative (QUAN) results?

The quantitative results gave us information that all students were pleased to be a member of the research school. The majority appreciated the activities, grants, and mentorship. Missing data were mainly those who recently had delivered their thesis and not yet participated in a PhD conference or made use of grants. This gave us information that the education was working, and again that activities, such as courses, conferences and networking were valued. However, adding the qualitative part gave more insight into their thoughts. Among these were the challenges in coming to Norway (a question not asked in the online survey), and that during the COVID pandemic they felt isolated and wished for more support. They also asked for support in the future and the transition from being a student to getting a job, and they wished access to the research school’s activities also after they had finished. The board of the school could later act upon these requests. Page 10   

7.Conclusion: Your conclusion should match the findings and explain well why this research was important/viable and what is it contribution in academia. Please rethink of the summary which match well as this looks like misplaced.

Thank you for pointing this out. We have now rephrased the conclusion and connected it to the aim of the study and the main aim of the education in the research school. Page 10

8.Reference section: Are not in APA format-Not even alphabetically written

In the guidelines from IJERPH it is recommended the EndNote references system which we have used. In this system, the references appear in the order they are referred to in the main text. We prefer and would like to keep the reference list as such. Please see a copy from the guidelines.

  • References: References must be numbered in order of appearance in the text (including table captions and figure legends) and listed individually at the end of the manuscript. We recommend preparing the references with a bibliography software package, such EndNote, ReferenceManager or Zotero to avoid typing mistakes and duplicated references. We encourage citations to data, computer code and other citable research material. If available online, you may use reference style 9. Page 11-12
  1. My comments: This research however can be good for scholarship if revisions are done extensively clarifying different components mentioned above.

Thank you very much for this encouraging comment. The manuscript has been sent to an authorized language controller before resubmission.

IRB

Copy of the Norwegian summary:

Vi vurderer at studien ikke er helseforskning, men annen type forskning. Prosjektet dreier

seg om synspunkter på aktiviteter i forskerskolen. Du har ikke planlagt å innhente eller

registrere helseopplysninger. Prosjektet er annen typen forskning (det vil si ikke medisinsk

eller helsefaglig forskning). Prosjektet er følgelig ikke omfattet av helseforskningslovens

saklige virkeområde, jf. helseforskningslovens §§ 2 og 4. Prosjektet kan derfor

gjennomføres og publiseres uten godkjenning fra REK.

Interpretation in English from Norwegian:

In our opinion, this study is not research on human health, but another kind of research. The project is about views on activities in a research school. You have not planned to ask for, or to register, health information. The project is another kind of research (meaning it is not medical or health research). Thus, the health research laws’ area, health research law §§ 2 and 4 does not apply to this project. The project can therefore be performed and published without approval from REK (= Regional Committees for Medical and Health Research Ethics).

Reviewer 2 Report

The aim of the study was to investigate previous PhD students’ views on the 14 Norwegian Research School of Global Health and its activities. A mixed methods study was used to determine demographic status, such as gender and work status, following the students’ PhD defense and their views on activities. By mixed method it seems that the researchers assumed the open questions used to explore perceptions of the activities etc. However, it is suggested to see the article “Bridging the Qualitative-Quantitative Divide: Experiences from Conducting a Mixed Methods Evaluation in the RUCAS Programme” that can be openly accessed. It is also suggested that the literature review should be more enriched with relevant to the research literature. Although, the results are based on a descriptive type of analysis, it is suggested to measure the reliability of the instrument used as well as its validity. It is also suggested to give more information on the composition of the sample and the methods of selecting the subjects.  It will also be interested if some hypotheses could be tested.

Author Response

Comments and Suggestions for Authors

Reviewer 2

Thank you very much for your valuable comments, which helped us to improve the manuscript! Please find our answers in red, below your comments.

The aim of the study was to investigate previous PhD students’ views on the 14 Norwegian Research School of Global Health and its activities.

Sorry, this is a misunderstanding, only one Norwegian Research School of Global Health has been evaluated. Page 1

A mixed methods study was used to determine demographic status, such as gender and work status, following the students’ PhD defense and their views on activities. By mixed method it seems that the researchers assumed the open questions used to explore perceptions of the activities etc. However, it is suggested to see the article “Bridging the Qualitative-Quantitative Divide: Experiences from Conducting a Mixed Methods Evaluation in the RUCAS Programme” that can be openly accessed.

Thank you for this relevant comment. We have read the suggested article and we acknowledge the authors’ view on the possibility and importance of combining a quantitative approach with a qualitative method to gain more understanding of the material. This was what we wanted in order to gain more knowledge about the material, the alumni as a group, and to determine if the quantitative and qualitative answers pointed in the same direction.

However, we have now omitted the expression ‘mixed method’ in describing the study design, both in the abstract and the main text, and we have used the terms quantitative and qualitative instead. We understand that the sample size of 60 alumni, with a response rate of 65% describes the material. No power-calculation was performed, no hypothesis testing, no comparison between groups, which makes it impossible to generalize the outcome. However, using some quantitative data, which were complemented by qualitative data, illustrated with quotes, gave us a deeper understanding of the previous students’ answers. Both the qualitative and quantitative data indicated similar experience which strengthened our conclusions. The activities were much valued, and in the qualitative part new information regard challenges, being in Norway also came forward. Page 2,5

It is also suggested that the literature review should be more enriched with relevant to the research literature.

We agree with the reviewer, we have included 7 more references from the literature. More about necessity to evaluate a postgraduate program, to ascertain what is needed for a better outcome is included. In the Introduction we refer to a study which concludes that there is insufficient information available to allow for meta-analysis of post graduate schools (Scott-Little, C et al. et al 2002). We found one study which evaluated a global health program, however for medical trainees, which is not directly comparable with our study group (Morgan J et al.2020). We also refer to e.g., Perkins et al 2020, who proposes that there is no "best approach" for evaluation and feedback. However, a multimodal approach can be effective and aid in guiding programs. The Sustainable Development Goals (SDGs) are referred to, and Ewing’s study in the Discussion regarding isolation during pandemic, which is accordance with ours. Page 2.3,4,9

Although, the results are based on a descriptive type of analysis, it is suggested to measure the reliability of the instrument used as well as its validity.

We understand and acknowledge your question about validity and reliability test. This would have required multiple iterations of piloting, which we honestly admit that we did not do. We did not use any validated questionnaire, as not much has been done in evaluating questionnaires to students coming from different settings to another country for post graduate studies. We developed the questionnaire ourselves. As our study mainly describes the material and has no statistical analyses, we instead have more focus on the qualitative data and its rigorous analysis.

It is also suggested to give more information on the composition of the sample and the methods of selecting the subjects. 

All 60 alumni who had already defended their thesis at a Norwegian university at the time the survey was carried out, were invited to answer an anonymous questionnaire online. Answers were received from 22 men and 17 women (Table 1).

It will also be interested if some hypotheses could be tested.

As there are only 60 alumni included in the study, and no power calculation was performed, we have not compared groups. However, performing a Chi2 test to see if there was a significant difference between men and women in being in a permanent job, vs in a temporary job or in a job seeking period, showed that men were more likely to have a job after the defense of their PhD thesis. The p value was .46995, < 0.5. But since these results are calculated from such a small material, we do not think it is relevant or appropriate to include this in the paper.  

Round 2

Reviewer 1 Report

Good job, now ! I am satisfied with improvements done so far.

Author Response

Thank you for your comments, we have further revise the manuscript based on Academic Editor's comments. Please find answers in the pdf file. 
